# Combined Effects of Botulinum Toxin Injection and Oral Appliance Therapy on Lower Facial Contouring: A Randomized Controlled Trial

**DOI:** 10.3390/jcm11144092

**Published:** 2022-07-14

**Authors:** YounJung Park, Sang Kyun Ku, Debora H. Lee, Seong Taek Kim

**Affiliations:** 1Department of Orofacial Pain and Oral Medicine, Yonsei University College of Dentistry, Seoul 03722, Korea; darkstar@yuhs.ac; 2Ye Dental Clinic, Seoul 07651, Korea; kosas75@naver.com; 3Department of Medicine, Irvine School of Medicine, University of California, Irvine, CA 92697, USA; deborhl1@hs.uci.edu

**Keywords:** botulinum toxins, oral appliance, lower facial contouring, masseter

## Abstract

(1) Background: Botulinum toxin (BoNT) injection is an esthetically effective and safe treatment for contouring the lower face. This study aimed to evaluate the combined effects of BoNT and supplementary oral appliance (OA) therapy on lower facial contouring. (2) Methods: We conducted a prospective randomized controlled trial from January 2015 to June 2016 at the Yonsei University Dental Hospital. Volunteers aged 20–45 years with masseter hypertrophy were randomly assigned to one of two groups: the non-OA group and the OA group. The non-OA group received BoNT injections alone, whereas the OA group received an OA in addition to BoNT injections. Changes in the bulkiest height of the lower face were evaluated by three-dimensional laser scanning before and 4, 8, 12, and 24 weeks after injections in both groups. (3) Results: In both groups, the bulkiest height reductions decreased, with a significant interaction between group (*p* = 0.046) and time (*p* < 0.001), although the overall reduction was at a similar level at 24 weeks. (4) Conclusions: The pattern of the bulkiest height reduction of the lower face after BoNT injection differed between standalone treatment and OA therapy, implying a normalizing effect of OA on masseter muscle activity.

## 1. Introduction

Masseter muscle hypertrophy is a clinical phenomenon that manifests as the enlargement of the masseter muscle and is accompanied by bruxism, clenching, facial pain, and functional impairment [1,2,3,4]. Varying degrees of success have been reported for various treatment options for masseteric hypertrophy with or without myogenous temporomandibular disorders (TMDs), ranging from simple pharmacotherapy, botulinum toxin (BoNT) injection, and oral appliance (OA) therapy to a more invasive surgical reduction in case of a bony protuberance of the mandibular angle [5,6,7].

After initially being described in 1994, BoNT injections have been widely used for the esthetic reshaping of the mandibular angle and to treat hypertrophy of masseter [8]. It ensures a temporary reduction in muscular activity in the injected muscles, generally beginning 2 weeks after the BoNT injection. In terms of the clinical effects, muscular atrophy usually lasts for 12–16 weeks (the atrophy phase), and the maximum reduction generally occurs at 12 weeks, after which muscular function begins to return (the recovery phase) [1,9]. Therefore, a booster injection may be recommended after the initial injection to maintain these effects.

OA is one of the most widely accepted methods for the treatment of TMDs, particularly those originating from the masticatory muscles. It is well known that OA intends to redistribute the occlusal forces, relax masticatory muscles, provide joint stabilization, and protect teeth [10]. As a result of OA, the masseter and temporalis muscle exhibit reduced electromyogram (EMG) activity, which correlates with reduced symptoms [11]. OA therapy has the possibility of reducing the dimensions of the masseter muscle [6].

Consequently, supplementary OA therapy following BoNT injection should maximize the effects of BoNT on lower face contouring by masseter muscle atrophy. In this randomized study, the effect of a single treatment with BoNT on masseter muscle hypertrophy was compared as a standalone treatment and in combination with OA therapy.

## 2. Materials and Methods

### 2.1. Participants

Volunteers aged 20–45 years were recruited through a recruitment notice between November 2015 and January 2016. The screening excluded volunteers having symptoms of pain-related TMD (diagnosed according to the diagnostic criteria for TMD [12]) such as arthralgia, myalgia, myofascial pain, and orofacial pain. Participants were included when they did not have a bony protuberance of the mandibular angle but had masseteric hypertrophy. Clinical examination was performed in all patients, including inspection and palpation of the masseter during rest and clenching. Based on palpation, more than mild masseter hypertrophy was included, such as bulging type I as described by Xie et al. [13].

The exclusion criteria were as follows: (1) pregnancy; (2) a history of any muscle dis-eases that can cause loss of muscle function, such as muscular dystrophy and derma-tomyositis; (3) receiving a BoNT injection, orthodontic treatment, or oral and maxillofacial plastic surgery within the previous 1 year to exclude any impact on the masseter. 

A total of 30 participants were treated at our institution between January 2015 and June 2016 (Table 1). This study was approved by the Institutional Review Board of Yonsei University Dental Hospital (IRB no. 2-2015-0037; approval date: 17 September 2015). All participants provided written informed consent to participate in this study. This study was performed in accordance with Tokyo’s (2004) revision of the 1975 Declaration of Helsinki.

### 2.2. Sample Size and Power

The sample size calculation was performed for a two-way repeated-measures analysis of variance (ANOVA) using G*Power 3.1 (Faul, Erdfelder, Lang, & Buchner, 2007). The effect size was set as medium (f = 0.25) for the OA and non-OA groups, each measured five times. The sample size was calculated to have a 90% power to reject the null hypothesis. The type I error probability associated with this test was 0.05. To achieve 90% power, 26 participants were required and considering a 10% follow-up loss, 15 participants were recruited for each study group.

### 2.3. Study Design 

This study was designed as a randomized, prospective controlled trial. Three researchers participated in the study: (1) the investigator assigned patients into two different groups, (2) the practitioner who administered the BoNT injection and OA delivery and check, and (3) the evaluator who measured using a three-dimensional laser scan.

At baseline, participants were randomly assigned to one of two groups by the investigator, who randomly allocated patients using a computer-generated randomization scheme (Microsoft Excel, Microsoft Corp., Redmond, WA, USA). The group assignment was kept in sealed envelopes that were accessible only to the investigator. The evaluator was blinded to the group assignment at all time points. In the non-OA group, 15 volunteers received BoNT injection alone, whereas, in the OA group, 15 volunteers applied OA every night for a period of 12 weeks after BoNT injection.

### 2.4. Intervention

#### 2.4.1. Information and Advice

At our clinic, all participants received information and advice concerning their lower facial contouring prior to any other treatment suggestion. This information and advice were provided by experienced clinicians.

#### 2.4.2. Botulinum Toxin Injection

A dose of 25 U of BoNT type A, letibotulinumtoxinA (Botulax^®^, Hugel Inc., Chuncheon, Korea) was injected bilaterally into each side of the masseter muscle using a 1-mL syringe with a 29-gauge, 0.5-inch-long needle. It was injected into two points 1 cm apart at the center of the lower third of the masseter muscle, as described in previous studies [14,15].

#### 2.4.3. Oral Appliance

An acrylic OA covering the maxillary anterior teeth, known as an anterior bite plane, was fabricated and adjusted individually for each subject (Figure 1). This appliance was applied during the night from weeks 1 to 12 after the BoNT injections in the OA group.

### 2.5. Measurement

The clinical effect of BoNT-A was evaluated by measuring the changes in the bulkiest height of the lower face on each side with a three-dimensional laser scan using a Vivid 9i^®^ laser scanner (Minolta, Tokyo, Japan) pre-injection and at 4, 8, 12, and 24 weeks post-injection. A single technical expert performed all scans, and each scan was merged into a single 3D facial image using image analysis software (Rapidform 2004; Inus Technology, Seoul, Korea). The forehead and parts of the glabella or nasal structures were used as reference areas to superimpose images, as described in previous studies, and pre- and post-injection images were superimposed on the regional best-fit method [16]. The maximum distance between the superimposed images at the bulkiest height of the lower face were bilaterally measured by superimposing the data acquired over time (Figure 2).

### 2.6. Statistical Analyses

The Mann–Whitney U test and chi-squared test were used to compare the demographic factors of the subjects. The paired *t*-test was used to compare the values measured on the right and left sides of 30 patients. Two-way repeated-measures ANOVA with group and time was used to evaluate the statistical significance of changes in the bulkiest height of the lower face over time and by the group. Mauchly’s sphericity test was used to verify the independent variables. When the probability value obtained in Mauchly’s sphericity test was less than 0.05, Greenhouse–Geisser (epsilon < 0.75) and Huynh–Feldt (epsilon > 0.75) corrections were used to modify the degrees of the freedom. Post-hoc tests were performed using an independent sample *t*-test with Bonferroni correction to compare the values measured at each time point between the two groups. Statistical significance was set at *p* < 0.05. All statistical analyses were performed using SPSS version 26.0 (IBM Corp., Armonk, NY, USA). Data are presented as means and standard deviations. 

## 3. Results 

### 3.1. Participants

In total, 30 volunteers (25 women and 5 men; average age, 31.8 years; age range, 22–41 years) were included in the study and randomly assigned to the non-OA or OA group. There were no dropouts in either group over a period of 24 weeks. The characteristics of the participants in the two groups are shown in Table 2. No statistically significant differences were found between the demographic data of the two groups (Table 2).

### 3.2. Summary Statistics

An evaluation using the paired *t*-test revealed that the changes in the bulkiest height of the lower face did not differ significantly between the left and right sides at each time point (*p* > 0.05). Hence, the average values obtained from the two sides were used for further statistical analyses. 

The means of the height change compared to the pre-injection state at each time point were acquired (Figure 3). 

The overall change in the bulkiest height of the lower face was at a similar level at 24 weeks. There were significant interactions between the time and group (with Greenhouse–Geisser correction, epsilon = 0.630, F = 2.966, *p* = 0.046) and a significant main effect of time (with Greenhouse–Geisser correction, epsilon = 0.630, F = 241.060, *p* < 0.001). In contrast, there was no significant main effect of the group (Table 3).

## 4. Discussion 

BoNT injection is esthetically effective and generally considered a safe option for contouring the lower face [1,2,9]. The effect of BoNT injections on lower face contouring does not appear immediately after the injection. Chemodenervation induces a reduction in muscle strength and causes slow muscle volume loss. The contouring effect resulting from BoNT injection appears clinically after neuromodulator-induced muscle atrophy. It is known that BoNT injections only change muscle thickness, not subcutaneous thickness [17]. Despite these changes in the target muscle, the result of lower face contouring depends clinically on the interrelation of complicated variables such as the amount of subcutaneous fat, skin laxity, degree of use of the masticatory muscles in daily life, and the amount of masseter muscle hypertrophy, and the individual’s anatomy. This course of muscular atrophy by BoNT is a reversible phenomenon, with partial recovery over 12–24 weeks, as presented in this study [1,18]. The masseteric muscular function returns with resynthesized neuromuscular synapses as a result of the sprouting of presynaptic axons [19].

Previously, clinical studies have reported changes from BoNT injection in the lower face using photographic measurements of the patient or ultrasonography or computed tomography scans to measure changes in the thickness of the masseter muscle [20,21]. However, recently developed 3D laser scanning provides more accurate and readily recognized measurements than other analysis tools [22,23]. This enables repeated imaging with no harmful effects and without direct contact with the patient’s face. This technology is currently being used to evaluate the efficacy of BoNTs [24]. This study analyzed changes in the bulkiest height of the lower face using 3D laser scanning, and the differences were determined after superimposition.

The masseter height of the bulkiest part was reduced in both groups, however, the effect was better in the non-OA group during the atrophy phase. In the OA group, the gradient of bulkiest height recovery was lower, indicating that the recovery rate was slower. This finding regarding the bulkiest height after wearing OA is probably attributable to the normalizing effect of OA, as suggested by Kurita et al. [25]. This means that the occlusal loads at the higher level decreased, whereas those at the lower level increased with the use of OA. The use of OA can normalize the occlusal force and the activity of the masseter muscle. Therefore, the effect of OA may change the pattern by altering occlusal function and stimulating the muscles of mastication in a manner that first limits the amount of muscular atrophy up to week 12 and then adapts to a baseline level during the recovery phase. With this normalizing effect, it is plausible to expect a late effect of OA that may slow the masseter muscle recovery thus, lasting the toxin effect and reducing the BoNT injection frequency. To confirm this possibility, it is necessary to add another group of people applying OA after the muscular atrophy phase following BoNT injection. Therefore, it would be interesting to focus on applying OA after the 12th week in the future.

The findings of this study should be interpreted considering several limitations. First, the major limitation of this study was the small sample size. Second, to better understand the combined effects of BoNT and OA on the recovery phase for lower face contouring, follow-up results of 3D laser measurements from a minimum of 12 to 18 months are needed. Third, the effectiveness of the anterior bite plane and full-arch occlusal splint, when combined with BoNT injections should be compared in future studies Fourth, since only one technical expert performed, merged, and superimposed the scans, unfortunately, intra-operator reproducibility was not considered in this study. Finally, the lack of control for sleep bruxism or daytime oral parafunction may have influenced our results. Despite these limitations, this is the first prospective study to evaluate the combined effects of BoNT and supplementary OA therapy on lower facial contouring. Further studies with larger numbers, a longer assessment period, and different application times of an OA after BoNT injection will provide more data to optimize the clinical effects of BoNT on lower facial contouring.

## 5. Conclusions

The bulkiest height pattern of the lower face after BoNT injection differed between the standalone treatment and the combination with OA therapy, implying a normalizing effect of OA on masseter muscle activity, although the overall reduction was at a similar level at 24 weeks.

## Figures and Tables

**Figure 1 jcm-11-04092-f001:**
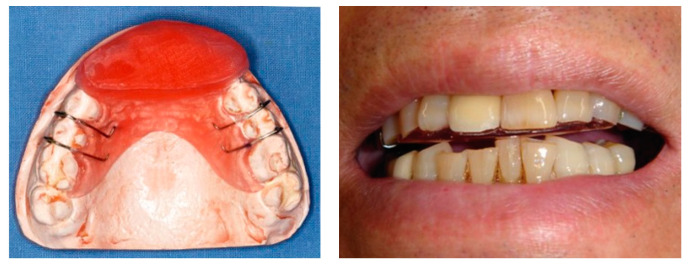
Oral appliance covering the maxillary anterior teeth.

**Figure 2 jcm-11-04092-f002:**
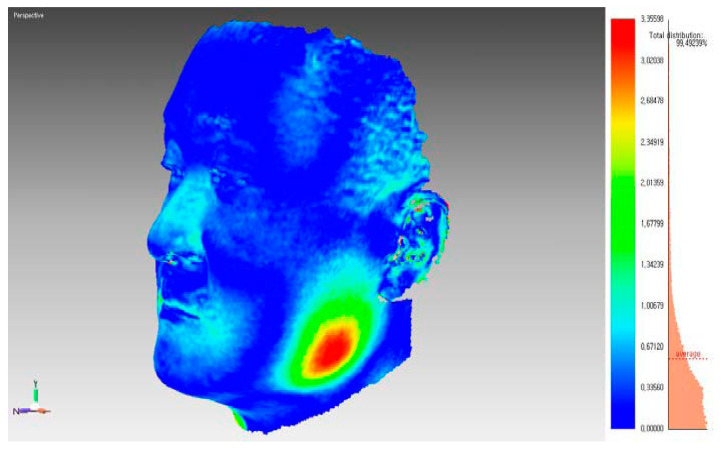
The bulkiest height of the lower face was measured by superimposing three-dimensional (3D) facial images.

**Figure 3 jcm-11-04092-f003:**
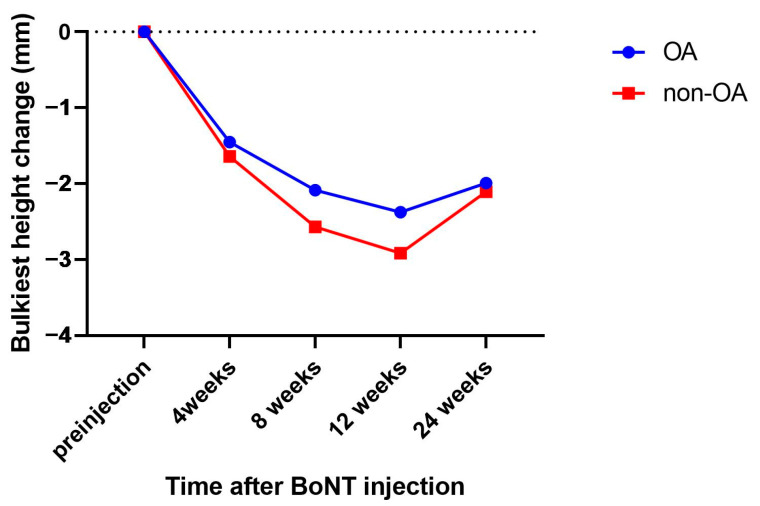
Changes in the bulkiest height of the lower face. OA, oral appliance; BoNT, botulinum toxin.

**Table 1 jcm-11-04092-t001:** Group overviews.

Participants	Non-OA Group (*n* = 15)		OA Group (*n* = 15)	
	Sex	Age (years)	Sex	Age (years)
1	M	22	F	25
2	F	26	F	26
3	F	27	F	26
4	F	27	F	27
5	M	28	M	28
6	M	29	F	32
7	F	29	F	32
8	M	30	F	33
9	F	31	F	34
10	F	32	F	35
11	F	33	F	38
12	F	33	F	39
13	F	34	F	40
14	F	39	F	40
15	F	39	F	41

OA, oral appliance; M, male; F, female.

**Table 2 jcm-11-04092-t002:** Demographic data of the participants.

Variables	Non-OA Group (*n* = 15)	OA Group (*n* = 15)	*p*	
Age (years)	30.0 [28.0–33.0]	33.0 [28.0–38.0]	0.169	Mann–Whitney U test
Sex			0.142	Chi-squared test
Female	14 (93.3)	11 (73.33)		
Male	1 (6.67)	4 (26.67)		

Values are presented as median [interquartile range] or *n* (%). OA, oral appliance.

**Table 3 jcm-11-04092-t003:** Mean change and standard deviations in the bulkiest height (mm) of the lower face at each time point. A minus sign indicates a reduction in measurement.

		Non-OA Group (*n* = 15)	OA Group (*n* = 15)
Time after BoNT injection	4 weeks	−1.78 ± 0.66	−1.45 ± 0.71
	8 weeks	−2.71 ± 0.77	−2.09 ± 0.73
	12 weeks	−2.99 ± 0.73	−2.37 ± 0.78
	24 weeks	−2.17 ± 0.79	−1.99 ± 0.73
*p*	Time	<0.001 *	
	Group	0.155	
	Time × group	0.046 *	

Main effects and interactions were tested using two-way repeated-measures analysis of variance (ANOVA). * *p* < 0.05. BoNT, botulinum toxin; OA, oral appliance.

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
