# Peer review of "Combined Effects of Botulinum Toxin Injection and Oral Appliance Therapy on Lower Facial Contouring: A Randomized Controlled Trial"

_jcm, 2022, doi:10.3390/jcm11144092_

Round 1
Reviewer 1 Report
Dear Authors
Thank you for your interesting article describing the benefit of the additional use of a bite plate after Botox Infiltration.
There are some issues that would enhance the value of your article
1. The article should be read and corrected by a native English-speaking person with scientific knowledge
2. The article could benefit from a list of the participants, age, weight, TMD score
3. Was weight considered as an influencing factor?
4. How was the presence or not of muscular hypertrophy measured, what criteria was used, a list will add to the value of the article
5. The abstract should state the goal of the research
6. The conclusion in the abstract is not clear
7. The literature list is not complete and is heavily balanced towards self-citation (in some instances rightfully so, but clearly overpowered)
Author Response
Response to Reviewer 1 Comments
I would like to express my appreciation for your advice.
We agreed with your comments and introduced corrections where required in the manuscript.
I look forward to your response and hope that the revised manuscript is suitable for publication in your journal.
Yours sincerely,
Seong Taek Kim
Point 1: The article should be read and corrected by a native English-speaking person with scientific knowledge
Response 1: The language was revised as per the reviewer’s recommendation.
Point 2: The article could benefit from a list of the participants, age, weight, TMD score.
Response 2: We agree with the reviewer’s opinion, so the list of the participants, age was added as Table 1. Unfortunately, weight was not considered in this study. Since participants were included after screening for the presence of pain-related TMD (diagnosed according to the diagnostic criteria for TMD), there was no participants who complain pain-related TMD symptoms.
Point 3: Was weight considered as an influencing factor?
Response 3: Unfortunately, weight was not considered as an influencing factor in this study. Further studies with weight will provide more data to find out the clinical effects of BoNT and oral appliance on lower facial contouring.
Point 4: How was the presence or not of muscular hypertrophy measured, what criteria was used, a list will add to the value of the article
Response 4: Clinical examination was performed in all patients, including inspection and palpation of the masseter on resting and clenching. More than mild masseter hypertrophy, bulging type I described by Xie et al(Xie, Y.; Zhou, J.; Li, H.; Cheng, C.; Herrler, T.; Li, Q. Classification of masseter hypertrophy for tailored botulinum toxin type a treatment. Plast Reconstr. Surg. 2014, 134, 209e-218e.), based on palpation was included.
The sentence explaining the measurement of muscular hypertrophy has been revised as follows:
Clinical examination was performed in all patients, including inspection and palpation of the masseter during rest and clenching. Based on palpation, more than mild masseter hypertrophy was included, such as bulging type I as described by Xie et al [14].
Point 5: The abstract should state the goal of the research
Response 5: The background in the abstract included the purpose of the research as follows:
(1) Background: Botulinum toxin (BoNT) injection is an esthetically effective and safe treatment for contouring the lower face. This study aimed to evaluate the combined effects of BoNT and sup-plementary oral appliance (OA) therapy on lower facial contouring.
Point 6: The conclusion in the abstract is not clear
Response 6: We agree with the reviewer’s opinion, so the conclusion in the abstract has been revised as follows:
(4) Conclusions: The bulkiest height pattern of the lower face after BoNT injection differed between standalone treatment and OA therapy, implying a normalizing effect of OA on masseter muscle activity.
Point 7: The literature list is not complete and is heavily balanced towards self-citation (in some instances rightfully so, but clearly overpowered)
Response 7: The literature list was revised as per the reviewer’s recommendation.
Please see the attachment (invoice for editing).

Reviewer 2 Report
I read with great interest the manuscript entitled "Combination Effects of Botulinum Toxin Injection and Oral Appliance therapy on Lower Facial Contouring".
The authors should respond to the following points:
- Authors should expand on the background of the study in the introduction. Authors should justify the need for this study.
- Authors should further describe the diagnostic criteria for temporomandibular disorders.
- What were the inclusion and exclusion criteria based on age?
- Why were patients who underwent plastic surgery in general excluded, and were patients who underwent plastic surgery, e.g. body plastic surgery, included?
- Why was one year considered as the exclusion time for participants undergoing plastic surgery?
- The authors should describe more precisely the inclusion and exclusion criteria of the study.
- When was patient recruitment performed?
- When was the study conducted?
- On the basis of what criteria was this type of oral appliance designed?
- The authors describe in the manuscript that they found no statistically significant differences in the demographics of the two groups, are there no statistically significant differences in the gender distribution of the groups?
- What are the limitations of this study?
- The conclusions are not based on the results of the study.
- The literature references used in this study are very limited.
Author Response
Response to Reviewer 2 Comments
I would like to express my appreciation for your advice.
We agreed with your comments and introduced corrections where required in the manuscript.
I look forward to your response and hope that the revised manuscript is suitable for publication in your journal.
Yours sincerely,
Seong Taek Kim
Point 1: Authors should expand on the background of the study in the introduction. Authors should justify the need for this study.
Response 1: The introduction was revised as per the reviewer’s recommendation.
Point 2: Authors should further describe the diagnostic criteria for temporomandibular disorders.
Response 2: We agree with the reviewer’s opinion, so the diagnostic criteria for temporomandibular disorders has been revised as follows:
After screening for the presence of pain-related TMD (diagnosed according to the diagnos-tic criteria for TMD [13]) such as arthralgia, myalgia, myofascial pain, and orofacial pain, participants were included
Point 3: What were the inclusion and exclusion criteria based on age? When was patient recruitment performed?
Response 3: The sentence about volunteer recruitment has been revised as follows:
Volunteers aged 20–45 years were recruited through a recruitment notice between November 2015 and January 2016.
Point 4: Why were patients who underwent plastic surgery in general excluded, and were patients who underwent plastic surgery, e.g. body plastic surgery, included? Why was one year considered as the exclusion time for participants undergoing plastic surgery?
The authors should describe more precisely the inclusion and exclusion criteria of the study.
Response 4: Patients who underwent plastic srugery only on oal and maxillofacial area within the previous 1 year were excluded to avoid the impact on the masseter.
The inclusion and exclusion criteria of the study has been revised as per the reviewer’s recommendation as follows:
After screening for the presence of pain-related TMD (diagnosed according to the diagnos-tic criteria for TMD [13]) such as arthralgia, myalgia, myofascial pain, and orofacial pain, participants were included when they did not have a bony protuberance of the mandibu-lar angle but had masseteric hypertrophy. Clinical examination was performed in all pa-tients, including inspection and palpation of the masseter during rest and clenching. Based on palpation, more than mild masseter hypertrophy was included, such as bulging type I as described by Xie et al [14].
The exclusion criteria were as follows: 1) pregnancy; 2) a history of any muscle dis-eases that can cause loss of muscle function, such as muscular dystrophy and der-ma-tomyositis; 3) receiving a BoNT injection, orthodontic treatment, or oral and maxillo-facial plastic surgery within the previous 1 year to exclude any impact on the masseter.
Point 5: When was the study conducted?
Response 5: The sentence about study method has been revised as follows:
A total of 30 participants were treated at our institution between January 2015 and June 2016.
Point 6: On the basis of what criteria was this type of oral appliance designed?
Response 6: An acrylic OA covering the maxillary anterior teeth, the so-called anterior bite plane, was applied to participants in this study. The use of this appliance seems to be as efficacious as the use of full-arch occlusal splint, as described by S. Tecco et al.( Tecco, S.; Caputi, S.; Tete, S.; Orsini, G.; Festa, F. Intra-articular and muscle symptoms and subjective relief during tmj internal derangement treatment with maxillary anterior repositioning splint or sved and mora splints: A comparison with untreated control subjects. Cranio 2006, 24, 119-129.) The possibility of adverse occlusal effects with this type of appliance only occurs with continuous and long-term use(Klasser, G.D.; Greene, C.S. Oral appliances in the management of temporomandibular disorders. Oral Surg. Oral Med. Oral Pathol. Oral Radiol. Endod. 2009, 107, 212-223.). Since the appliance was applied during the night from weeks 1 to 12 after the BoNT injections in OA group in this study, we used this appliance.
Point 7: The authors describe in the manuscript that they found no statistically significant differences in the demographics of the two groups, are there no statistically significant differences in the gender distribution of the groups?
Response 7: There no statistically significant differences in the gender distribution of the groups (p =0.142) using Chi-squared test.
Point 8: What are the limitations of this study?
Response 8: . The major limitation is the small sample size. To better understand combined effects of BoNT and OA on the recovery phase for lower face contouring, the follow-up results of 3D laser measurement from a minimum of 12 months to 18 months is needed. The effectiveness of the anterior bite plane and full-arch occlusal splint when combined with BoNT injection should be compared in further studies. The lack of control for sleep bruxism or daytime oral parafunctions might have influenced our results though and is a limitation of our study.
Point 9: The conclusions are not based on the results of the study.
Response 9: We agree with the reviewer’s opinion, so the conclusion has been revised as follows:
The bulkiest height pattern of the lower face after BoNT injection differed between the standalone treatment and the combination with OA therapy, implying a normalizing ef-fect of OA on masseter muscle activity.
Point 10: The literature references used in this study are very limited.
Response 10: The literature references were revised as per the reviewer’s recommendation.
Please see the attachment (invoice for editing).

Round 2
Reviewer 2 Report
The authors have significantly improved the manuscript in line with the points made in the first revision.
Congratulations on the work!
